# Beyond Node-Centric Modeling: Sketching Signed Networks with Simplicial Complexes

**Wei Wu[1]    Xuan Tan[1]    Yan Peng[1]    Ling Chen[2]    Fangfang Li[1,3]***    **Chuan Luo[4]**

[1]School of Computer Science and Engineering, Central South University
[2]Australian Artificial Intelligence Institute, University of Technology Sydney
[3]Xiangjiang Laboratory
[4]School of Software, Beihang University
william.third.wu@gmail.com   {tanxuan, pengyan, lifangfang}@csu.edu.cn
ling.chen@uts.edu.au   chuanluo@buaa.edu.cn

## Abstract

Signed networks can reflect more complex connections through positive and negative edges, and cost-effective signed network sketching can significantly benefit an important link sign prediction task in the era of big data. Existing signed network embedding algorithms mainly learn node representation in the Graph Neural Network (GNN) framework with the balance theory. However, the node-wise representation learning methods either limit the representational power because they primarily rely on node pairwise relationship in the network, or suffer from severe efficiency issues. Recent research has explored simplicial complexes to capture higher-order interactions and integrated them into GNN frameworks. Motivated by that, we propose EdgeSketch+, a simple and effective edge embedding algorithm beyond traditional node-centric modeling that directly represents edges as low-dimensional vectors without transitioning from node embeddings. The proposed approach maintains a good balance between accuracy and efficiency by exploiting the Locality Sensitive Hashing (LSH) technique to swiftly capture the higher-order information derived from the simplicial complex in a manner of no learning processes. Experiments show that EdgeSketch+ matches state-of-the-art accuracy while significantly reducing runtime, achieving speedups of up to $546.07\times$ compared to GNN-based methods[2].

## 1   Introduction

*Signed networks* provide a richer and more nuanced description of the connections than unsigned networks by differentiating positive and negative relationships. This enables critical applications like analyzing social cohesion through friend/enemy ties and improving recommender systems via positive/negative feedback. Consequently, a significant task is *link sign prediction*, which aims to predict the sign of the edge in a signed network.

Current signed network embedding methods primarily use node-wise representation learning in Graph Neural Networks (GNNs) [1–9], incorporating the balance theory (i.e., a friend of my friends is my friend; a enemy of my enemies is my friend) [10, 11]. Although the node-wise signed network embedding approaches have achieved great success thanks to the balance theory, they either are trapped into the special circumstances of the edges such as isomorphic proximity or incur severe

---

*Corresponding author.
[2]We have released the source code and the datasets in `https://github.com/AIandBD/EdgeSketchplus`.

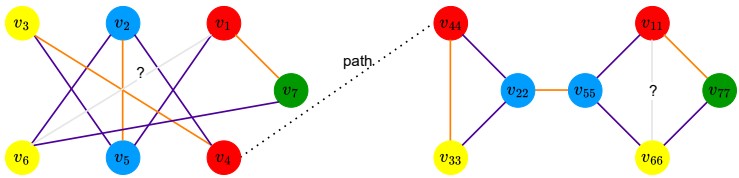

Figure 1: In a signed network, orange and purple edges represent positive and negative signs, respectively. Balance theory suggests $v_1$ and $v_6$ are enemies, while $v_{11}$ and $v_{66}$ are indistinct. Edges $(v_1, v_6)$ and $(v_{11}, v_{66})$ have isomorphic proximity in node-to-node interactions but differ in a simplicial complex context.

efficiency issues from neural network training. For example, in Figure 1, the balance theory cannot identify the edge $(v_{11}, v_{66})$ but $(v_1, v_6)$ (i.e., enemies). Meanwhile, unsigned network embedding methods, including both node embedding [12–18] and edge embedding [19–23], lack mechanisms to capture signed network properties, resulting in performance degradation in dealing with signed networks.

Furthermore, a great many graph embedding methods concentrate on the node-wise interaction relationship, but real-world networks are increasingly complicated, rendering node-to-node relationships incompetent to delineate higher-order interaction relationships in the networks. For example, social network analysis emphasizes friend circles rather than individual pairs [24]. As shown in Figure 1, edges $(v_1, v_6)$ and $(v_{11}, v_{66})$, which are isomorphic at the node level, differ in higher-order structures — $(v_{11}, v_{66})$ lies in two triangles while $(v_1, v_6)$ is in only one. Recent advancements in GNNs have incorporated simplicial complexes [25–35], which exhibit the powerful representation capability by capturing the complicated, higher-order substructures in the network, yet these approaches demand huge computational cost due to massive parameter training.

By contrast, randomization offers high efficiency in algorithm design by avoiding laborious learning processes. A well-established framework, Locality-Sensitive Hashing (LSH) [36], supported by a solid theoretical foundation, embeds similar data objects into the same location in a low-dimensional space with higher probability than dissimilar ones by randomized hash functions, preserving their relative distances. To our knowledge, most graph hashing methods [37–51] efficiently represent graphs or nodes in reduced dimensions in a non-learning fashion, though [47, 51] enhance graph classification by converting node-based graphs into simplicial complex-based ones. As this integration remains underexplored, we focus on advancing signed network sketching using LSH and simplicial complexes, aiming for robust theoretical guarantees and strong experimental results.

In this paper, we seek to strike a good balance between accuracy and efficiency of sketching the signed network by leveraging the LSH technique to capture higher-order interaction relationship from the simplicial complex. We first propose a novel edge structure distance measure tailored for the edge surrounding, which is represented via the convolution operation on the edge itself (i.e., 1-simplex) and all its direct neighborhoods containing the adjacent edges (i.e., 1-simplexes), nodes (i.e., 0-simplexes) and triangles (i.e., 2-simplexes) under the simplicial complex, preserving intricate structural information beyond traditional node-wise neighborhoods (See Appendix C). Subsequently, our cost-effective algorithm beyond node-centric modeling, EdgeSketch+, iteratively sketches edge surroundings into low-dimensional binary vectors without learning, while maintaining low computational overhead. Additionally, by doubling the binary representation length, EdgeSketch+ enables integration with linear models, effectively addressing SVM kernel storage issues in large-scale scenarios. In summary, the main contributions of our work are as follows:

- To our knowledge, this is the first work introducing simplicial complexes to signed networks, significantly improving embedding efficiency and link sign prediction.

- We present a novel signed network sketching model called EdgeSketch+, which effectively and efficiently represents each signed edge by adopting the LSH technique to capture the higher-order interaction relationship from the simplicial complex.

- The experimental results show that our proposed EdgeSketch+ algorithm enjoys good performance comparable to the state-of-the-art competitors, while remarkably reducing computational expenses, for example, running up to 546.07× faster than the GNN-based signed network embedding methods.

## 2 Related Works

**Graph Embedding with Simplicial Complex.** In order to cope with the complicated interaction in the graph, the researchers have explored the use of simplicial complex in the GNN framework for applications such as trajectory prediction [34, 29], visual inspection [31] and missing data imputation [33]. MPSN [35] and SaNN [25] perform graph classification in the message passing mechanism with the simplicial complex, while N$k$F [32] outperforms the existing message passing neural networks [52–54] by adopting differential $k$-forms on simplicial complexes to augment geometric interpretability. SGAT [26] implements node embedding by injecting simplicial complexes into heterogeneous message passing networks, and SCRaW1 [27] learns node representation by random walks on higher-order simplices. Furthermore, SCN [30] and EdgeRWSE [28] extend node (i.e., 0-simplex) embedding to edge (i.e., 1-simplex) embedding on message passing and random walks, respectively; the latter additionally preserves information from local/global positions of the nodes and local/global structures in the network based on edge-level random walks.

**Signed Network Embedding.** The signed networks assign the edges with a richer and more nuanced description such that the positive and negative edges are distinguished. In particular, link sign prediction is an important task unique to the signed networks and thus has attracted much attention from researchers. To our knowledge, most of the existing signed network embedding algorithms [1–4, 7, 5, 6, 55, 8] are node-wise representation under the balance theory [10, 11]. They usually apply a certain GNN method (e.g., GCN [53], GAT [52], GCL [56], etc.) to the signed networks based on the balance theory. RSGNN [1] aims to improve the robustness of the signed GNN methods by reducing the affect of noisy edges. Furthermore, GS-GNN [2], generalizing the balance theory to the $k$-group theory [57, 58], describes the node relationship with all the groups as the global information and models the positive and negative edges as the local information. SGA [59] addresses graph sparsity and unbalanced triangle issues through structure-aware edge manipulation and curriculum learning. In addition, many node embedding algorithms [12–17] are initially used for the classic downstream tasks in the traditional networks, e.g., node classification and link prediction. By contrast, only a few works [19–23] focus on the edge embedding. These methods commonly lack the mechanism of capturing the unique properties of signed networks. Whether for signed networks or traditional networks, the above network embedding algorithms mainly rely on random walks, matrix factorization and graph neural networks, which pose challenges in terms of efficiency.

**Graph Hashing.** Although the GNN framework has achieved great progress, it suffers from huge computational resource and memory storage due to dramatic parameter training. To this end, the researchers have resorted to the Locality Sensitive Hashing (LSH) technique, which can efficiently conduct the high-dimensional data similarity estimation by exploiting a family of random hash functions to map the similar data objects to the close and even the same data points represented as vectors, e.g., MinHash [60], SimHash [61], etc. Some graph hashing algorithms [37–42, 62, 48–50] aim to represent each node in the network as a compact hashcode. The core idea of the methods is to extract and sketch the subtrees rooted at each node as the corresponding node representation. Although the methods can further derive the edge representation, they cannot preserve the complete characteristics of the edges. By contrast, others [43–47, 51] represent the whole graph as a low-dimensional feature vector for graph classification. Particularly, the works [47, 51] improve the performance of the above LSH-based graph classification by extracting the simplicial complex and then converting the original node-based graphs into multi-dimensional simplex-based graphs. Unfortunately, the whole graph hashing methods are not applicable in our problem setting.

## 3 Preliminaries

### 3.1 Edge-based Signed Network Embedding

Given a signed network $g = (\mathcal{V}, \mathcal{E}^+ \cup \mathcal{E}^-)$, where $\mathcal{V}$ is a node set, $\mathcal{E}^+ \subseteq \mathcal{V} \times \mathcal{V}$ is a positive edge set and $\mathcal{E}^- \subseteq \mathcal{V} \times \mathcal{V}$ is a negative edge set. Note that $\mathcal{E}^+ \cap \mathcal{E}^- = \emptyset$ means that an edge $e$ is either positive or negative. Edge-based signed network embedding seeks to sketch each edge $e \in \mathcal{E}^+ \cup \mathcal{E}^-$ as a low-dimensional vector $\mathbf{x}_e \in \mathbb{R}^D$, which facilitates the link sign prediction task.

## 3.2 Simplicial Complex

Given a network $g = (\mathcal{V}, \mathcal{E})$, where $\mathcal{V}$ is a node set and $\mathcal{E}$ is an edge set, we call it $k$-simplex $\mathcal{S}_k = [v_0, v_1, \ldots, v_k]$ if a node subset $\{v_0, v_1, \ldots, v_k\} \subseteq \mathcal{V}$ forms a clique in $g$, for example, a 0-simplex is a node, a 1-simplex is an edge, a 2-simplex is a triangle, etc. Also, given a $k$-simplex $\mathcal{S}_k = [v_0, \ldots, v_k]$ and a $(k-1)$-simplex $\mathcal{S}_{k-1} = [v_0, \ldots, v_{k'-1}, v_{k'+1}, \ldots, v_k]$, $\mathcal{S}_k$ is a co-boundary adjacency and $\mathcal{S}_{k-1}$ is a boundary adjacency. Formally, the boundary operator is $\partial \mathcal{S}_k = \sum_{k'=0}^{k} (-1)^{k'} \mathcal{S}_{k-1} = \sum_{k'=0}^{k} (-1)^{k'} [v_0, \ldots, v_{k'-1}, v_{k'+1}, \ldots, v_k]$. Furthermore, a simplicial complex $C$ is a collection of different dimensional simplexes in the network. We let $K$ be the largest dimension of any simplex in $C$, and $C_k$ be the subset composed of all the $k$-simplexes. We encode the adjacency relationship between $(k-1)$-simplexes and $k$-simplexes as +-, where the columns represent the corresponding coefficients of the $(k-1)$-simplexes in the above-mentioned boundary operator of all the $k$-simplexes. Consequently, the Hodge Laplacian of a $K$-dimensional simplicial complex is defined as

$$
\begin{aligned}
\mathbf{L}_0 &= \mathbf{B}_1 \mathbf{B}_1^\top \\
\mathbf{L}_k &= \mathbf{B}_k^\top \mathbf{B}_k + \mathbf{B}_{k+1} \mathbf{B}_{k+1}^\top, k \in \{1, 2, \ldots, K-1\} \\
\mathbf{L}_K &= \mathbf{B}_K^\top \mathbf{B}_K
\end{aligned}
\tag{1}
$$

where each row and each column of $\mathbf{L}_k$ denotes a $k$-simplex, $k \in \{0, 1, \ldots, K\}$. Similarly, $\mathbf{L}_k$ describes the adjacency relationship between any two $k$-simplexes. Note that an entry of $\mathbf{B}_k / \mathbf{L}_k$ is 0 if the two corresponding simplexes are not adjacent; otherwise, it is non-zero. The simplicial complex bridges geometric topology with algebraic topology, which further establishes a connection between graph analysis and topology. In a word, the simplicial complex assists in preserving higher-order interaction relationship in the network.

## 3.3 Locality Sensitive Hashing

An efficient framework for data sketching and similarity preserving backed by solid theoretical foundation is Locality-Sensitive Hashing (LSH) [36], which sketches the similar data objects as the same hashcode with a higher probability than the dissimilar ones by a family of randomized hash functions.

**Definition 1** (Locality Sensitive Hashing). A family $\mathcal{H}$ of randomized functions is called $(d_1, d_2, p_1, p_2)$-sensitive if for two data objects $\mathbf{p}$ and $\mathbf{q}$, and $\forall h \in \mathcal{H}$:

- If $dist(\mathbf{p}, \mathbf{q}) \leq d_1$, then $\Pr(h(\mathbf{p}) = h(\mathbf{q})) \geq p_1$;
- If $dist(\mathbf{p}, \mathbf{q}) \geq d_2$, then $\Pr(h(\mathbf{p}) = h(\mathbf{q})) \leq p_2$,

where $d_1 < d_2$ under a certain distance measure $dist$ and $0 \leq p_2 < p_1 \leq 1$.

## 4 Signed Network Sketching

In this section, we propose a novel signed network sketching model called EdgeSketch+ beyond node-centric modeling, which employs the simplicial complex to capture the higher-order interaction relationship in the signed network and then efficiently implements edge embedding in the LSH framework. The relevant definitions are provided in Appendix C.

### 4.1 The EdgeSketch+ Algorithm

We outline the proposed EdgeSketch+ method in Algorithm 1. The input consists of a signed network $g = (\mathcal{V}, \mathcal{E}^+ \cup \mathcal{E}^-)$, the number of dimensions for edge embedding $D$, the number of iterations $R$, and $R$ sets of $3 \times D$ random vectors $\{\mathbf{h}_k^{(d,r)}\}_{k=0,d=1,r=1}^{2,D,R}$ corresponding to 0-, 1- and 2-simplexes, respectively. The output returns each edge embedding at the $R$th iteration as the final edge embedding.

First, Algorithm 1 extracts all the simplexes (Line 1), and establishes the simplex-based adjacency matrices and the 1-dimensional Hodge Laplacian (Lines 2-3). Subsequently, Algorithm 1 proceeds in an iterative way: we represent all the extracted simplexes by the convolution operation on the simplex-based adjacency matrices and the 1-dimensional Hodge Laplacian (Lines 5-7); further, we

**Algorithm 1** The EdgeSketch+ Algorithm

---

**Input:** $g = (\mathcal{V}, \mathcal{E}^+ \cup \mathcal{E}^-)$, $D$, $R$, $\{\mathbf{h}_k^{(d,r)}\}_{k=0,d=1,r=1}^{2,D,R}$

**Output:** $\{\mathbf{x}_{e_i}^{(R)}\}_{i=1}^{|\mathcal{E}^+ \cup \mathcal{E}^-|}$

1: Extract all 0-, 1- and 2-simplexes in $g$
2: Build the simplex-based adjacency matrices $\mathbf{B}_1$ and $\mathbf{B}_2$
3: Build the 1-dimensional Hodge Laplacian $\mathbf{L}_1$
4: **for** $r = 1, \ldots, R$ **do**
5: $\quad [\mathbf{x}_{n_1}^{(r-1)}; \ldots; \mathbf{x}_{n_{N_0}}^{(r-1)}] \leftarrow \text{Conv}_{\mathbf{B}_1}([\mathbf{x}_{n_1}^{(r-1)}; \ldots; \mathbf{x}_{n_{N_0}}^{(r-1)}])$
6: $\quad [\mathbf{x}_{t_1}^{(r-1)}; \ldots; \mathbf{x}_{t_{N_2}}^{(r-1)}] \leftarrow \text{Conv}_{\mathbf{B}_2}([\mathbf{x}_{t_1}^{(r-1)}; \ldots; \mathbf{x}_{t_{N_2}}^{(r-1)}])$
7: $\quad [\mathbf{x}_{e_1}^{(r-1)}; \ldots; \mathbf{x}_{e_{N_1}}^{(r-1)}] \leftarrow \text{Conv}_{\mathbf{L}_1}([\mathbf{x}_{e_1}^{(r-1)}; \ldots; \mathbf{x}_{e_{N_1}}^{(r-1)}])$
8: $\quad$ **for** $i = 1, \ldots, N_1$ **do**
9: $\quad\quad \mathbf{x}_{e_i}^{(r)} \leftarrow \text{AGG}(\{\mathbf{h}_1^{(d,r)}\mathbf{x}_{e_i}^{(r-1)}\}_{d=1}^D, \{\mathbf{h}_0^{(d,r)}\mathbf{x}_n^{(r-1)} | \mathbf{x}_n^{(r-1)} \text{is boundary adjacent on } \mathbf{B}_1\}_{d=1}^D,$
$\quad\quad\quad \{\mathbf{h}_2^{(d,r)}\mathbf{x}_t^{(r-1)} | \mathbf{x}_t^{(r-1)} \text{is co-boundary adjacent on } \mathbf{B}_2\}_{d=1}^D, \{\mathbf{h}_1^{(d,r)}\mathbf{x}_e^{(r-1)} | \mathbf{x}_e^{(r-1)} \text{is adjacent on } \mathbf{L}_1\}_{d=1}^D)$

10: $\quad\quad \mathbf{x}_{e_i}^{(r)} \leftarrow \text{sgn}(\mathbf{x}_{e_i}^{(r)})$
11: $\quad$ **end for**
12: **end for**

---

aggregate (e.g., sum) each edge surrounding in the LSH framework such that the direct neighborhood information is effectively and efficiently perserved (Lines 8-11). Accordingly, the resulting edge embeddings are from the last iteration. We illustrate the overall procedure of the proposed EdgeSketch+ approach in Figure 4 of Appendix B.

The random vectors are produced off-line only once and shared globally. Therefore, there is no demand for high-performance hardware such as GPUs. As a consequence, the proposed EdgeSketch+ algorithm remarkably saves computation cost.

### 4.2 Theoretical Analysis

In this section, we theoretically analyze the proposed EdgeSketch+ algorithm.

#### 4.2.1 EdgeSketch+ belongs to an LSH family

**Proposition 1**. Let $e_i$ and $e_j$ be two edges, respectively, and $D$ be the size of their sketchings. Algorithm 1 is $(d_1, d_2, 1 - \frac{d_1}{D}, 1 - \frac{d_2}{D})$-sensitive for some large $D$:

- if $dist(e_i, e_j) \leq d_1$, then $\Pr(h(e_i) = h(e_j)) \geq 1 - \frac{d_1}{D}$,
- if $dist(e_i, e_j) \geq d_2$, then $\Pr(h(e_i) = h(e_j)) \leq 1 - \frac{d_2}{D}$,

where $0 \leq d_1 < d_2 \leq 1$.

**Proof**. The Hamming distance between two edge sketchings varies from 0 to some large $D$ in the case of $D$-dimensional sketches, i.e., $dist(e_i, e_j) \in [0, D]$. Furthermore, we can convert the distance into the probability that they share the same hashcode, i.e.,

$$\Pr(h(e_i) = h(e_j)) = 1 - \frac{dist(e_i, e_j)}{D} \in [0, 1]. \tag{2}$$

#### 4.2.2 EdgeSketch+ produces a Hash Kernel

**Definition 2** (Hash Kernel). Given any two edges, $e_i$ and $e_j$, Algorithm 1 generates the $D$-dimensional edge sketching under $R$ iterations, $\mathbf{x}_{e_i}^{(R)}$ and $\mathbf{x}_{e_j}^{(R)}$, respectively. We define the hash kernel, $\kappa_{EdgeSketch+}$, between $e_i$ and $e_j$ as

$$\kappa_{EdgeSketch+}(e_i, e_j) = \kappa(\mathbf{x}_{e_i}^{(R)}, \mathbf{x}_{e_j}^{(R)}) = \frac{1}{D}\sum_{d=1}^{D} \mathbb{1}(x_{e_i,d}^{(R)} = x_{e_j,d}^{(R)}). \tag{3}$$

**Theorem 1**. The hash kernel matrix $\mathbf{K} \in \mathbb{R}^{N \times N}$, as defined in Eq. (3), is positive definite.

**Proof.** We rewrite Eq. (3) as

$$\kappa_{EdgeSketch+}(e_i, e_j) = \frac{1}{D} \sum_{d=1}^{D} \left( \mathbb{1}(x_{i,d} = 1) \times \mathbb{1}(x_{j,d} = 1) + \mathbb{1}(x_{i,d} = 0) \times \mathbb{1}(x_{j,d} = 0) \right)$$

$$= \frac{1}{\sqrt{D}} \times \left( \mathbb{1}(x_{i,1} = 1), \cdots, \mathbb{1}(x_{i,D} = 1), \mathbb{1}(x_{i,1} = 0), \cdots, \mathbb{1}(x_{i,D} = 0) \right) \cdot \quad (4)$$

$$\frac{1}{\sqrt{D}} \times \left( \mathbb{1}(x_{j,1} = 1), \cdots, \mathbb{1}(x_{j,D} = 1), \mathbb{1}(x_{j,1} = 0), \cdots, \mathbb{1}(x_{j,D} = 0) \right)^{\top}.$$

The value $\kappa_{EdgeSketch+}(e_i, e_j)$ can be interpreted as the inner product of two vectors, each having a dimensionality of $2D$. As a result, $\mathbf{K}$ can be expressed as $\mathbf{K} = \mathbf{MM}^{\top}$, where $\mathbf{M} \in \mathbb{R}^{N \times 2D}$.

### 4.2.3 Concentration

We define the theoretical similarity between two edges, $e_i$ and $e_j$, as

$$Sim(e_i, e_j) = \Pr(h(e_i) = h(e_j)) \quad = \lim_{D \to +\infty} \frac{1}{D} \sum_{d=1}^{D} \mathbb{1}(x_{e_i,d}^{(R)} = x_{e_j,d}^{(R)}). \quad (5)$$

Then, we can show the highly-concentrated estimator of $Sim(e_i, e_j)$.

**Theorem 2.** Given two edges $e_i$ and $e_j$, Algorithm 1 generates the corresponding $D$-dimensional edge sketching $\mathbf{x}_{e_i}^{(R)}$ and $\mathbf{x}_{e_j}^{(R)}$ under $R$ iterations. $\forall \epsilon > 0$, the probability of deviation between the kernel $\kappa_{EdgeSketch+}(e_i, e_j)$ and their real similarity $Sim(e_i, e_j)$ is bounded by the following inequality

$$\Pr[|\kappa_{EdgeSketch+}(e_i, e_j) - Sim(e_i, e_j)| \geq \epsilon] \leq 2 \exp(-2D\epsilon^2). \quad (6)$$

**Proof.** Let $X_d^{(R)}$ be a Bernoulli random variable, which takes $X_d^{(R)} = 1$ with the probability of $Sim(e_i, e_j)$. Furthermore, considering $D$ i.i.d Bernoulli random variables, $X_1^{(R)}, X_2^{(R)}, \cdots, X_D^{(R)}$, we have $\overline{X}^{(R)} = \frac{1}{D} \sum_{d=1}^{D} X_d^{(R)}$ and $\mathbb{E}[\overline{X}^{(R)}] = Sim(e_i, e_j)$. Consequently, we can directly use the Hoeffding bound [63] on the mean of the variables,

$$\Pr[|\kappa_{EdgeSketch+}(e_i, e_j) - Sim(e_i, e_j)| \geq \epsilon] = \Pr[|\overline{X}^{(r)} - \mathbb{E}[\overline{X}^{(r)}]| \geq \epsilon]$$
$$\leq 2 \exp(-2D\epsilon^2). \quad (7)$$

### 4.2.4 Time Complexity

Let $\overline{v}$ be the average degree of the network, $R$ be the number of iterations, and $D$ be the number of dimensions for edge embedding. We denote the number of 0-simplexes, 1-simplexes and 2-simplexes as $N_0$, $N_1$ and $N_2$, respectively. In the real scenarios, the networks are commonly sparse, which means that $O(N_0) = O(N_1) = O(N_2)$ [64, 65]. Consequently, it takes $O(N_0)$, $O(N_1)$ and $O(\overline{v}N_1)$ to extract 0-simplexes, 1-simplexes and 2-simplexes, respectively. Next, Algorithm 1 spends $O(2N_1)$, $O(3N_2)$ and $O(N_1^2)$ in generating the sparse matrices, $\mathbf{B}_1$, $\mathbf{B}_2$ and $\mathbf{L}_1$, respectively. In the outer for loop, it requires $O(N_1D)$ (Line 5), $O(N_1D)$ (Line 6), $O(N_1D)$ (Line 7) and $O(\overline{v}N_1D^2)$ (Lines 8-11). Consequently, the total time complexity is $O(N_1^2 + \overline{v}N_1D^2R)$.

Additionally, we analyze the algorithm's representational power and space complexity in Appendix D.1 and Appendix D.2, respectively.

## 4.3 Incorporation with Linear Learning Models

As in Eq. (3), the $D$-dimensional edge embedding derives nonlinear kernels. However, the explicit kernel computation is definitely accompanied with a precomputed Gram matrix with the size being the square of the number of training instances. Fortunately, the inner product in Eq. (4) indicates that the vectorized representation can be mapped into an inner product space by negation and concatenation, which remarkably speedups the training and testing processes without compromise to classification accuracy by the linear learning models such as logistic regression. Consequently, this approach

effectively addresses the issue of SVM training for large-scale problems, particularly when the training dataset exceeds available memory capacity. Taking the edge embedding vector $[1, 0, 1, 0, 0]$ as an example, Eq. (4) transforms it into a 10-dimensional feature vector, i.e., $\frac{1}{\sqrt{5}} \times [1, 0, 1, 0, 0, \underline{0}, \underline{1}, \underline{0}, \underline{1}, \underline{1}]$, where the underlined entries negate the corresponding bits in the original vector. Consequently, the concatenated vectors can be utilized by linear solvers, eliminating the need to store large-scale kernel matrices.

# 5 Experimental Results

In this section, we conduct the extensive experiments to evaluate the performance of the proposed EdgeSketch+ method. All dataset information is provided in Table 6 of Appendix E, where "BTC-$\alpha$" stands for Bitcoin-alpha, "BTC-O" for Bitcoin-OTC, "Slash." for Slashdot, and "Epin." for Epinions. For details on baselines and experimental settings, please refer to Appendix F.

## 5.1 Link Sign Prediction Results

Table 1: Link sign prediction performance results.

| Datasets | Metrics | RSGNN | SDGNN | SNEA | SGCL | SiGAT | GSGNN +SGA | EdgeRWSE | TER | edge2vec | MPSketch | EdgeSketch+ |
|---|---|---|---|---|---|---|---|---|---|---|---|---|
| | Binary-F1 | 0.9475 | 0.9714 | 0.9272 | 0.9740 | 0.9706 | 0.9603 | 0.9659 | 0.9704 | 0.9686 | 0.9705 | **0.9780** |
| | Accuracy | 0.9051 | 0.9458 | 0.8729 | 0.9508 | 0.9438 | 0.9266 | 0.9340 | 0.9435 | 0.9263 | 0.9433 | **0.9581** |
| BTC-$\alpha$ | AUC | 0.8467 | 0.8799 | 0.7863 | **0.9101** | 0.8874 | 0.8911 | 0.5314 | 0.8579 | 0.6246 | 0.8485 | 0.8829 |
| | Macro-F1 | 0.5874 | 0.7294 | 0.7119 | 0.7501 | 0.6758 | 0.7373 | 0.4836 | 0.6770 | 0.4892 | 0.5874 | **0.7687** |
| | Runtime (s) | 332.81 | 779.81 | 315.71 | 2204.68 | 267.83 | 176.69 | 558.67 | 94.67 | 4081.36 | 121.23 | **4.03** |
| | Binary-F1 | 0.9533 | 0.9623 | 0.9138 | 0.9666 | 0.9600 | 0.9655 | 0.9467 | 0.9627 | 0.9617 | 0.9605 | **0.9681** |
| | Accuracy | 0.9172 | 0.9315 | 0.8585 | 0.9384 | 0.9265 | 0.9209 | 0.8987 | 0.9327 | 0.9263 | 0.9315 | **0.9413** |
| BTC-O | AUC | 0.8127 | 0.9004 | 0.8032 | **0.9152** | 0.8953 | 0.8987 | 0.5314 | 0.8579 | 0.6132 | 0.8127 | 0.8695 |
| | Macro-F1 | 0.6096 | 0.7850 | 0.7652 | 0.7782 | 0.7473 | 0.7533 | 0.4836 | 0.6770 | 0.4892 | 0.6096 | **0.8137** |
| | Runtime (s) | 441.78 | 1394.25 | 430.51 | 2930.11 | 716.56 | 173.65 | 1744.71 | 125.64 | 4560.63 | 460.23 | **7.81** |
| | Binary-F1 | - | - | 0.8737 | - | 0.9072 | 0.8672 | - | 0.8741 | - | - | **0.9209** |
| | Accuracy | - | - | 0.8162 | - | 0.8519 | 0.7982 | - | 0.7795 | - | - | **0.8729** |
| Slash. | AUC | - | - | 0.7942 | - | 0.8864 | 0.8222 | - | 0.7129 | - | - | **0.8914** |
| | Macro-F1 | - | - | 0.7665 | - | 0.7697 | 0.7232 | - | 0.4710 | - | - | **0.7995** |
| | Runtime (s) | OOT | OOT | 3119.04 | OOT | 11182.62 | 1326.12 | OOM | 652.34 | OOT | OOM | **140.44** |
| | Binary-F1 | - | - | 0.9255 | - | 0.9579 | 0.9523 | - | 0.9206 | - | - | **0.9659** |
| | Accuracy | - | - | 0.8784 | - | 0.9269 | 0.9188 | - | 0.8530 | - | - | **0.9405** |
| Epin. | AUC | - | - | 0.8240 | - | **0.9248** | 0.8914 | - | 0.6340 | - | - | 0.9217 |
| | Macro-F1 | - | - | 0.7954 | - | 0.8354 | 0.8222 | - | 0.4601 | - | - | **0.8658** |
| | Runtime (s) | OOM | OOT | 4180.38 | OOT | 30651.82 | 1220.41 | OOM | 1069.47 | OOT | OOM | **273.24** |

OOM/OOT mean that the methods run out of memory/time.

Table 1 shows the experimental results. Clearly, our proposed EdgeSketch+ algorithm competes very well with all the competitors tailored for signed network embedding (i.e., RSGNN, SDGNN, SNEA, SGCL, SiGAT and GSGNN+SGA) – it performs best in most cases, with an average AUC lag of only 0.0253, compared to the top-performing baselines, which reflects that the proposed EdgeSketch+ algorithm has achieved strong precision and recall for the minority class but struggles to maintain balanced performance across all the thresholds due to the influence of the majority class; also, it runs fastest on all the datasets, achieving up to 546.07× speedup[3]. Particularly, we would like to note that some GNN-based algorithms suffer from huge time or memory consumption on the two largest datasets, i.e., Slash. and Epin.[4] This reveals that the idea of integrating the simplicial complex into the LSH framework is very promising in efficiently recognizing the confusing patterns of signed networks with no data distribution fitting and further benefiting the quality of edge-based signed network embedding — our proposed EdgeSketch+ algorithm judges the edges more accurately with the aid of simplicial complexes. In the Appendix G, we analyze the experimental results about the edge embedding algorithms (i.e., EdgeRWSE, TER and edge2vec) and the LSH-based method (i.e., MPSketch) in more details.

---

[3]The ratios exclude the out-of-time and out-of-memory scenarios.

[4]In the original paper [1], RSGNN ran only on the sampled Slash. and Epin.

## 5.2 Ablation Study

The proposed EdgeSketch+ algorithm preserves higher-order interaction relationship from the simplicial complex in the signed network by the LSH technique. Therefore, we conduct the ablation analysis w.r.t. the simplicies (i.e., 0-simplex, 1-simplex and 2-simplex) involved in the proposed EdgeSketch+ algorithm. To this end, we have the following variants,

- *EdgeSketch+-$L_1$* only utilizes the Hodge Laplacian $\mathbf{L}_1$ by removing the boundary adjacency matrix involving 0-simplexes, $\mathbf{B}_1$, and the co-boundary adjacency matrix involving 2-simplexes, $\mathbf{B}_2$, simultaneously;

- *EdgeSketch+-$I^\top I$* replaces $\mathbf{L}_1$ of EdgeSketch+-$L_1$ with $\mathbf{I}^\top\mathbf{I}$, which operates the incidence matrix $\mathbf{I}$ in Eq. (1). Any entry of $\mathbf{I}_1$ is 1 if the node is incident upon the edge; it is 0, otherwise.

We adopt the same parameters for the two variants for a fair comparison. Table 2 reports the experimental results in terms of $D = 300$ and $R = 1$. EdgeSketch+-$\mathbf{I}^\top\mathbf{I}$, with no simplicies involved, performs worst, which indicates that the simplicial complex, as the higher-order abstraction, definitely facilitates signed network sketching by preserving the higher-order interaction relationship. Furthermore, we observe the wider gap between EdgeSketch+ and EdgeSketch+-$\mathbf{L}_1$, which implies that the 0-simplexes and 2-simplexes involved in the edge surrounding are indispensable and the higher-order direct neighborhood information remarkably benefits the final edge embedding. Overall, the performance shows an obvious increasing trend as more simplicies are captured, i.e., from just incidence information to the edge surrounding.

Table 2: Ablation study of EdgeSketch+ w.r.t. the simplicial complex.

| Datasets | Metrics | EdgeSketch+ -$\mathbf{I}^\top\mathbf{I}$ | EdgeSketch+ -$\mathbf{L}_1$ | EdgeSketch+ (orignal) |
|---|---|---|---|---|
| BTC-$\alpha$ | Binary-F1 | 0.9690 | 0.9696 | **0.9780** |
|  | Accuracy | 0.9404 | 0.9415 | **0.9581** |
|  | AUC | 0.7624 | 0.7786 | **0.8829** |
|  | Macro-F1 | 0.6120 | 0.6182 | **0.7687** |
| BTC-O | Binary-F1 | 0.9556 | 0.9574 | **0.9681** |
|  | Accuracy | 0.9173 | 0.9205 | **0.9413** |
|  | AUC | 0.7594 | 0.7667 | **0.8695** |
|  | Macro-F1 | 0.6789 | 0.6841 | **0.8137** |
| Slash. | Binary-F1 | 0.8990 | 0.8996 | **0.9209** |
|  | Accuracy | 0.8343 | 0.8351 | **0.8729** |
|  | AUC | 0.8120 | 0.8133 | **0.8914** |
|  | Macro-F1 | 0.7194 | 0.7216 | **0.7995** |
| Epin. | Binary-F1 | 0.9456 | 0.9459 | **0.9659** |
|  | Accuracy | 0.9038 | 0.9044 | **0.9405** |
|  | AUC | 0.8441 | 0.8462 | **0.9217** |
|  | Macro-F1 | 0.7664 | 0.7676 | **0.8658** |

## 5.3 Scalability

As shown in Section 4.2.4, the time and space complexities of the proposed EdgeSketch+ algorithm are both polynomial w.r.t. the number of edges, and thus we verify its scalability w.r.t. the the network property on a set of random, million-scale networks (i.e., $|\mathcal{V}| = 10^6$) generated by the Erdos-Renyi model [66]. Figure 2 reports the results in terms of memory consumption and runtime in case of $R = 2$ and $D = 300$. The embedding time of EdgeSketch+ displays the slow quadratic growth, owing to $\bar{v}, D, D^2, R \ll N_1$, and the memory consumption is empirically linear thanks to

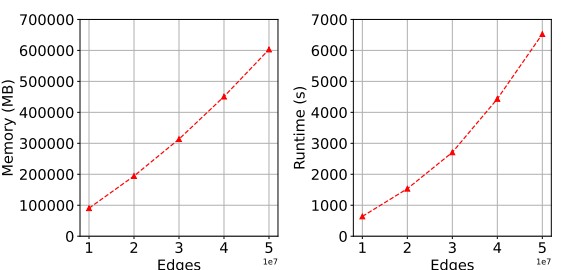

Figure 2: Scalability of EdgeSketch+ w.r.t. the number of edges.

$DR \ll N_1$. Particularly, our proposed probabilistic model produces representation for tens of millions of edges in less than 2 hours and 600GB on the million-scale networks. The empirical results match the theoretical complexities. Therefore, our proposed EdgeSketch+ model shows good potential in efficiently sketching large-scale signed networks.

## 5.4 Hyper-parameter Sensitivity

The proposed EdgeSketch+ algorithm has two parameters, i.e., the embedding dimension $D$ and the number of iterations $R$. We explore how these two parameters affect link sign prediction performance

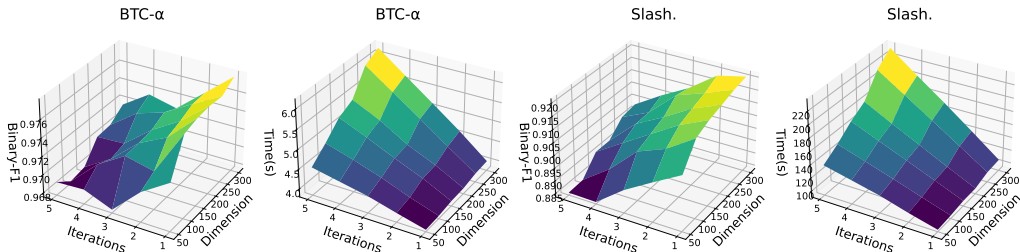

Figure 3: Hyper-parameter sensitivity in EdgeSketch+ in link sign prediction w.r.t. the embedding dimension $D$ and the number of iterations $R$.

in effectiveness and runtime. We exhibit the results in terms of binary-f1 and end-to-end runtime on BTC-$\alpha$ and Slash. in Figure 3.

It is evident that the accuracy performance is closely influenced by the embedding dimension $D$ and the number of iterations $R$. Generally speaking, accuracy exhibits fluctuations as $D$ and $R$ increase on BTC-$\alpha$; whereas we observe a clear convex surface on Slash. Particularly, EdgeSketch+ performs best in the case of the largest $D = 300$ and the smallest $R = 1$. This result highlights two key insights: a larger $D$ captures more meaningful information and the simplicial complex, as a higher-order abstraction, retains sufficient information with a small $R$. Conversely, increasing $R$ excessively would cause feature diffusion across simplexes, leading to the loss of local features. The end-to-end runtime demonstrates an empirical linear relationship with $R$, even though the classification time is considered. This implies that the embedding process theoretically linear to $R$ dominates the whole link sign prediction task, and further EdgeSketch+ generates high-quality embedding vectors that are easy to classify.

## 5.5 Effectiveness of Edge Structure Distance

We define the edge structure distance based on the Hamming distance to describe the similarity between the edges of the signed networks in Appendix C. In order to verify its effectiveness, we compare with the common Cosine distance. Specifically, we precompute the Cosine kernel based on cosine distance and then feed it into SVM.

Table 3 shows the comparison results on BTC-$\alpha$ and BTC-O. Clearly, our proposed edge similarity based on the Hamming distance is superior to the Cosine similarity in the LSH family.

Table 3: Hamming distance *vs.* Cosine distance.

| Datasets | Metrics | EdgeSketch+ -Cosine | EdgeSketch+ (orignal) |
|---|---|---|---|
| BTC-$\alpha$ | Binary-F1 | 0.9742 | **0.9780** |
| | Accuracy | 0.9505 | **0.9581** |
| | AUC | 0.7329 | **0.8829** |
| | Macro-F1 | 0.6509 | **0.7687** |
| BTC-O | Binary-F1 | 0.9616 | **0.9681** |
| | Accuracy | 0.9285 | **0.9413** |
| | AUC | 0.7158 | **0.8695** |
| | Macro-F1 | 0.5481 | **0.8137** |

## 5.6 Necessity of Edge Embedding

In order to demonstrate that the edge embedding as feature design definitely fosters the good performance, we directly feed the initialized edge vectors into the logistic regression classifier.

Table 4 shows the comparison results on BTC-$\alpha$ and BTC-O. In the scenario where the same classifier model is adopted, our proposed edge embedding model performs much better than the naive method based on the only initialized edge vectors, which shows that the edge embedding effectively captures the higher-order structure information. By contrast, the naive method captures only the information of the edges themselves. Particularly, the AUC values of 0.5000 from the

Table 4: Importance of the Edge Embedding.

| Datasets | Metrics | Naive | EdgeSketch+ |
|---|---|---|---|
| BTC-$\alpha$ | Binary-F1 | 0.9698 | **0.9780** |
| | Accuracy | 0.9415 | **0.9581** |
| | AUC | 0.5000 | **0.8829** |
| | Macro-F1 | 0.4830 | **0.7687** |
| BTC-O | Binary-F1 | 0.9436 | **0.9681** |
| | Accuracy | 0.8933 | **0.9413** |
| | AUC | 0.5000 | **0.8695** |
| | Macro-F1 | 0.4786 | **0.8137** |

naive method imply that good feature design is very necessary; otherwise, the classifier performs no better than random guessing.

## 6 Conclusion

In this paper, we propose a simple and very speedy signed network sketching model dubbed EdgeSketch+ beyond node-centric modeling, which can capture the higher-order information derived from the simplicial complex with more representational capability than the traditional node-to-node interactive relationship. The approach adopts the LSH technique to avoid substantial parameter training, while preserving as much structural information as possible. Furthermore, we provide the theoretical guarantees for the accuracy and the complexity of the edge representation. We conduct the extensive experiments of EdgeSketch+ and a collection of state-of-the-art methods. We evaluate its performance in terms of accuracy and runtime on a number of signed network datasets. The experimental results show that our proposed EdgeSketch+ method achieves the competitive accuracy performance with dramatically reduced runtime, which makes it more practical in the era of big data.

## Acknowledgments

This work was supported by Open Project of Xiangjiang Laboratory (No.25XJ03020), National Natural Science Foundation of China (No. 62302528, 62522201, 62172449, 62202025, 72374070), Hunan Provincial Natural Science Foundation of China (2022JJ3021,2025JJ20071), Beijing Natural Science Foundation (No. L241050), Young Elite Scientist Sponsorship Program by CAST (No. YESS20230566), CCF-Huawei Populus Grove Fund CCF-Huawei (No. CCF-HuaweiFM2024005), High Performance Computing Center of Central South University, and Fundamental Research Fund Project of Beihang University.

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

# Appendix

## A   Notation Table

The notations are summarized in Table 5.

Table 5: Notations

| Notation | Description |
|---|---|
| $g$ | Signed network where edges have positive or negative signs |
| $\mathcal{V}$ | Set of nodes in the signed network |
| $\mathcal{E}$ | Set of edges in the signed network |
| $v$ | A node in the signed network |
| $e$ | An edge in the signed network, typically a 1-simplex |
| $\mathcal{S}_k$ | A $k$-simplex |
| $\mathbf{x}_\mathcal{S}$ | Embedding of simplex $\mathcal{S}$ |
| $\mathcal{C}$ | A simplicial complex |
| $\mathrm{B}_i$ | Boundary adjacency matrix for $i$-simplices |
| $\mathrm{L}_i$ | $i$-dimensional Hodge Laplacian |
| $\mathcal{H}$ | A family of randomized functions |
| $\mathbf{h}$ | Random vector |
| $\kappa$ | Hash kernel |

## B   The EdgeSketch+ Algorithm

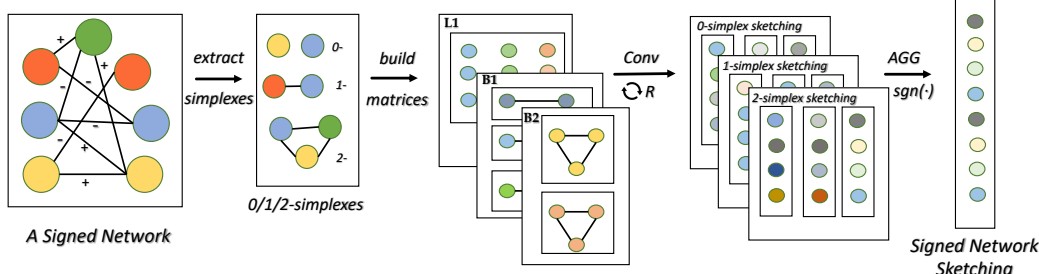

Figure 4: The overall procedure of the proposed EdgeSketch+ algorithm.

## C   Definitions

**Definition 3** (Edge Surrounding). The edge surrounding centers around the edge itself (i.e., 1-simplex), with its direct neighborhood from the viewpoint of the simplicial complex, including the adjacent edges (i.e., 1-simplexes), nodes (i.e., 0-simplexes) and triangles (i.e., 2-simplexes). We can represent the edge surrounding via the convolution operation on the simplex-based adjacency matrices $\mathbf{B}_1$, $\mathbf{B}_2$ and the 1-dimensional Hodge Laplacian $\mathbf{L}_1$, which effectively preserves the direct neighborhood information.

This definition extends the traditional neighborhood relationship in graph mining, which is limited to the same-level, node-centric adjacency relationship.

**Definition 4** (Edge Sketching). The edge $e$'s sketching under $R$ iterations is a $D$-dimensional binary vector, $\mathbf{x}_e^{(R)} = [x_{e,1}^{(R)}, x_{e,2}^{(R)}, \cdots, x_{e,D}^{(R)}]$, which is defined in terms of the edge surrounding projected into $R$ sets of $3 \times D$ random vectors in an iterative manner. The vector element $x_{e,d}^{(R)} = 1$ if the $d$th projection is non-negative, and 0 otherwise.

The projection operation is actually a linear combination of three kinds of simplexes from the edge surrounding on three random vectors, which is mathematically equivalent to the edge surrounding holistically projected into a target random vector.

**Definition 5** (Edge Structure Distance). Given any two edges, $e_i$ and $e_j$, Algorithm 1 generates the $D$-dimensional edge sketching under $R$ iterations, $\mathbf{x}_{e_i}^{(R)}$ and $\mathbf{x}_{e_j}^{(R)}$, respectively. We define the edge structure distance as the Hamming distance between their sketchings, i.e.,

$$dist(e_i, e_j) = dist(\mathbf{x}_{e_i}^{(R)}, \mathbf{x}_{e_j}^{(R)}) = \sum_{d=1}^{D} \mathbb{1}(x_{e_i,d}^{(R)} \neq x_{e_j,d}^{(R)}), \tag{8}$$

where $x_{e_i,d}^{(R)} \in \{0, 1\}$, and $\mathbb{1}(state) = 1$, if $state$ is true, and $\mathbb{1}(state) = 0$, otherwise.

The distance represents the number of the target random vectors that lie in the same direction with exactly one of two edge surroundings.

## D  Theoretical Analysis

### D.1  Representational Power

The work [54] has demonstrated that the WL isomorphism test [67] has the maximal representational power to discriminate the substructures in the graph because the test ensures that different substructures is embedded into distinct locations in the feature space.

Motivated by the LSH framework, we can quantize the representational power of our proposed EdgeSketch+ algorithm from a probabilistic perspective. Assuming that $\hbar^{(r)}$ denotes the $r$-th sketching process of the proposed probabilistic model, we have the probabilistic expression as follows

$$\Pr(\hbar^{(r)}(\mathbf{x}_{e_i}^{(r-1)}) = \hbar^{(r)}(\mathbf{x}_{e_j}^{(r-1)})) \quad = \lim_{D \to +\infty} \frac{1}{D} \sum_{d=1}^{D} \mathbb{1}(x_{e_i,d}^{(r)} = x_{e_j,d}^{(r)}), \tag{9}$$

where $r = 1, 2, \cdots, R$. Formally, Eq. (9) illustrates that the proposed EdgeSketch+ model could represent two substructures in the graph as the same feature vector with the probability of their theoretical similarity.

### D.2  Space complexity

EdgeSketch+ requires storing the simplex-based adjacency matrices (i.e., $O(2N_1)$ for $\mathbf{B}_1$ and $O(3N_2)$ for $\mathbf{B}_2$) and the Hodge Laplacian (i.e., $O(\bar{\nu}N_1)$ for $\mathbf{L}_1$) as well as the embedding for 0-simplexes (i.e., $O(N_0 D)$), 1-simplexes (i.e., $O(N_1 D)$) and 2-simplexes (i.e., $O(N_2 D)$); also, the random vectors need $O(D^2 R)$. Therefore, the space complexity is practically $O(N_1 D + D^2 R)$.

## E  Datasets

We implement the link sign prediction task on the following signed networks from the SNAP group[5].

- *Bitcoin-alpha* is a who-trust-whom Bitcoin-Alpha trading network, where the users give trust or distrust tags to others for the sake of security.
- *Bitcoin-OTC* is a who-trust-whom Bitcoin-OTC trading network, where the users give trust or distrust tags to others for the sake of security.
- *Slashdot* is a news website that covers technology, science and culture. The users are allowed to tag each other as friends or foes.
- *Epinions* is a who-trust-whom online social network from the consumer review site. The users decide whether to trust each other.

Particularly, for Bitcoin-alpha and Bitcoin-OTC, users rate others negatively (i.e., $\{-10, -9, \cdots, -2, -1\}$) or positively (i.e., $\{10, 9, \cdots, 2, 1\}$); we consider the negative scores

---

[5]https://snap.stanford.edu/data/index.html

Table 6: Summary of the signed network datasets.

| Datasets | $|\mathcal{V}|$ | $|\mathcal{E}^+|$ | $|\mathcal{E}^-|$ | |triangles| | $|\bar{\nu}|$ |
|---|---|---|---|---|---|
| Bitcoin-alpha | 3,783 | 22,650 | 1,536 | 22,153 | 12.79 |
| Bitcoin-OTC | 5,881 | 32,029 | 3,563 | 33,493 | 12.10 |
| Slashdot | 82,144 | 425,072 | 124,130 | 579,565 | 13.37 |
| Epinions | 131,828 | 717,667 | 123,705 | 4,910,076 | 12.76 |

as the negative edges and the positive scores as the positive edges, as shown in [1, 3]. We initialize the node features with the positive in-degree, the negative in-degree, the positive out-degree and the negative out-degree; then, we initialize the features of high-dimensional simplexes (i.e., edges and triangles) by taking the sum of the node features. The above datasets are summarized in Table 6.

# F  Baselines and Experiment Setting

We compare the proposed EdgeSketch+ method with the following state-of-the-art approaches in the link sign prediction task[6].

- *RSGNN* [1] aims to denoise the signed network in the node representation learning process based on the balance theory.
- *SDGNN* [3] learns node representation by aggregating messages from different signed, social relations in the signed network based on the balance theory.
- *SNEA* [4] estimates the importance coefficients of node pairs via the self-attention mechanism when representing nodes in the signed network based on the balance theory.
- *SGCL* [6] learns node representation by applying the graph contrastive learning to signed networks based on the balance theory.
- *SiGAT* [7] learns node representation by applying the graph attention network to signed networks based on the balance theory.
- *GSGNN+SGA* [2, 59] combines GS-GNN's [2] latent group modeling with SGA's [59] augmentation to address graph sparsity and unbalanced triangles.
- *EdgeRWSE* [28] learns edge representation based on edge-level (i.e., 1-simplex) random walks with edge-level node position information supplemented.
- *TER* [20] exploits the randomized singular value decomposition [68] of edge transition matrices to obtain edge representation.
- *edge2vec* [19] utilizes deep autoencoders to capture local and global structure information of the embedded edges.
- *MPSketch* [39] generates the node embeddings by using the LSH technique to pass messages in the MPNN framework [69].

All the compared methods have their codes provided by their respective authors, and we adopt the recommended hyperparameters in their papers for the learning methods. Also, we conduct the experiments with different values of $R \in \{1, 2, 3, 4, 5\}$ for MPSketch and EdgeSketch+. Finally, we set $R = 1$ on Bitcoin-alpha and $R = 4$ on Bitcoin-OTC for MPSketch, which runs out of memory on Slashdot and Epinions; we set $R = 1$ on Bitcoin-alpha and $R = 1$ on Bitcoin-OTC, $R = 1$ on Slashdot and $R = 1$ on Epinions for EdgeSketch+. We set the embedding dimension $D$ as 300 and the cutoff time of 24 hours. All the experiments are conducted on Linux with 2.90GHz 128 Intel Xeon Platinum 8375C CPU, 1.5T RAM and NVIDIA A10 GPU (24GB RAM). When the experiments encounter out-of-memory errors on the GPU, we would switch the computational device to the CPU. Following the previous works [1, 2], we achieve the signed network embedding and then conduct the link sign prediction task by the logistic regression classifier for the methods which are not trained in an end-to-end manner. Particularly, for our proposed EdgeSketch+ algorithm, we expand the embedding vectors and then feed them into the logistic regression classifier, as shown in Section

---

[6]We do not report the results from [48, 55, 30] since the codes are not publicly available.

4.3; for MPSketch, we use the Hamming distance and SVM classifier recommended in the original paper[7]. Otherwise, we employ the end-to-end training. We report the mean of binary-f1, accuracy, auc, macro-f1 and end-to-end runtime in the 5-fold cross validation in Table 1. We provide the code in the attachment.

# G Analysis of Experimental Results

The proposed EdgeSketch+ model outperforms all the edge embedding algorithms (i.e., EdgeRWSE, TER and edge2vec). In spite of the simplicial complex involved, EdgeSketch+ is unsurprisingly superior to EdgeRWSE, because EdgeRWSE focuses on random walks on simplicial complexes, with the edge patterns among the signed triangles neglected, that is to say, the neighbors that share 0-simplexes is only considered; by contrast, EdgeSketch+ preserves such important information related to 2-simplexes to distinguish some difficult patterns under the balance theory. Besides, EdgeRWSE exhausts memory on Slashdot and Epinions, mainly due to the memory consumption required for walking between a large number of simplexes. TER, as an efficient and effective edge embedding method recently claimed by its authors, definitely performs more efficiently than the learning-based competitors by randomized singular value decomposition [68], but EdgeSketch+ goes towards more efficient and effective by leveraging the advantages of the LSH technique and the simplicial complex. Also, EdgeSketch+ clearly outperforms edge2vec in terms of effectiveness and efficiency, because edge2vec essentially preserves local and global proximities from the node-level perspective by deep autoencoders. This again illustrates that the simplicial complex is able to identify more patterns.

Also, we compare EdgeSketch+ against the most recent LSH-based node embedding algorithm, MPSketch, which derives edge embedding by concatenating two corresponding node embeddings. Evidently, MPSketch performs worst, and even runs out of memory on Slashdot and Epinions, because it cannot effectively capture the substructure features in the signed networks and more importantly, it embeds into the nonlinear Hamming space such that the precomputed kernel matrix is too big to fit in memory, which makes it difficult to scale. Although the two methods do not need learning, EdgeSketch+ still performs faster than MPSketch by two orders of magnitude. The main reasons are as follows: 1) MPSketch sketches a large, aggregated information pool, and even iterates more than EdgeSketch+ on Bitcoin-OTC capture sufficient information; 2) The SVM classifier training based on the Hamming kernel is time-consuming. This again implies that MPSketch has the limited capabilities of capturing the patterns in the signed networks.

In addition, we would like to note that all the competitors implicitly/explicitly represent the nodes or edges as 32/64-bit embedding vectors, while our proposed EdgeSketch+ algorithm only generates the binary vectors with nearly no sacrificing the result quality, which means that EdgeSketch+ is able to enjoy lower memory footprints than the compared methods in signed network embedding. Overall, our proposed EdgeSketch+ model achieves an excellent balance between accuracy and efficiency.

# H Limitations

This work has two main limitations. First, although EdgeSketch+ operates in an unsupervised manner without using edge signs during sketching, it relies on downstream classifiers that require labeled data. This setup may limit its applicability in domains where labeled signed edges are scarce. Second, the current approach leverages simplicial complexes constructed from triangle structures, which may miss relevant higher-order interactions not captured by 2-simplices. Extending the framework to incorporate higher-dimensional simplices could provide more comprehensive modeling, albeit at the cost of increased computational complexity.

---

[7]We also assess MPSketch by logistic regression, but it underperforms.

