# OpenReview forum: "Beyond Node-Centric Modeling: Sketching Signed Networks with Simplicial Complexes"
_NeurIPS.cc/2025/Conference — NeurIPS 2025 poster_

### Official Review · Reviewer_dXe3 · 2025-06-16

**Clarity:** 3
**Significance:** 4
**Originality:** 3
**Rating:** 5
**Confidence:** 5

**Summary:**

This paper introduces a novel and well-motivated edge-based signed network sketching method, which preserves higher-order relationship from simplicial complexes beyond the traditional node-centric modeling in the signed network. The method is elegant in its simplicity yet powerful in its expressiveness, marking a significant departure from traditional node-based embedding techniques. To my best knowledge, very few existing works have explored the use of higher-order topological structures in graphs, particularly for signed networks. The authors skillfully bridge the gap between traditional signed graph sketching and higher-order topology, demonstrating that capturing structures beyond pairwise interactions significantly enhances downstream performance, avoiding the issue from the balance theory. Furthermore, the Locality-Sensitive Hashing (LSH) framework helps to efficiently capture higher-order interactions without the computational burden of traditional GNNs. Particularly, integration with linear learning models guarantees real-world large-scale applications. Consequently, the method effectively balances performance and efficiency while maintaining strong representational capability. The work is both theoretically grounded and practically impactful, offering a scalable and versatile tool for signed network analysis.

**Questions:**

see the weaknesses

**Ethical Concerns:**

["NO or VERY MINOR ethics concerns only"]

**Final Justification:**

The authors have adequately addressed all of my concerns in their rebuttal. I am satisfied with their clarifications and maintain my positive evaluation. I recommend acceptance of the manuscript.

**Limitations:**

yes

**Quality:**

3

**Strengths And Weaknesses:**

Strengths:

1. The paper is well-written and easy to follow.
2. The idea of simplicial complexes as higher-order abstraction in signed network embedding is interesting. The method beyond the traditional node-centric scheme provides a new insight into signed network embedding.
3. Extensive experimental results across multiple datasets and baselines highlight the effectiveness, efficiency, scalability, and domain-agnostic nature of the proposed framework.
4. The paper provides solid theoretical analysis, including LSH-based theory, error bound, and complexity, with the performance guaranteed. Particularly, incorporation with linear models benefits large network application, avoiding the issue of SVM training, and also the experiments verify this point.

Weaknesses:

1. It would be better if the paper provides a notation table to clarify many symbols.
2. The authors empirically verify the necessity of edge embedding. The authors are encouraged to further demonstrate that the method effectively preserves the essential structural information of the data.
3. Section 4.2.3 theoritically analyzes the error bound. Empirical study is encouraged.

---

> ### Author Rebuttal · Authors · 2025-07-31
>
> *Q1: Notation table*
>
> A1: The notation table is as follows
> |Notation|Description|
> |-----------------|-------------|
> |$g$|Signed network where edges have positive or negative signs|
> |$\mathcal{V}$|Set of nodes in the signed network|
> |$\mathcal{E}$|Set of edges in the signed network|
> |$v$|A node in the signed network|
> |$e$|An edge in the signed network, typically a 1-simplex|
> |$\mathcal{S}_k$|A $k$-simplex|
> |$\mathbf{x}_{\mathcal{S}}$|Embedding of simplex $\mathcal{S}$|
> |$\mathcal{C}$|A simplicial complex|
> |$\mathrm{B}_i$|Boundary adjacency matrix for $i$-simplices|
> |$\mathrm{L}_i$|$i$-dimensional Hodge Laplacian|
> |$\mathcal{H}$|A family of randomized functions|
> |$\mathbf{h}$|Random vector|
> |$\kappa$|Hash kernel|
>
> *Q2: essential structural information*
>
> A2: To assess whether our edge embedding method preserves essential structural information, we perform a distributional similarity analysis between the embeddings before and after sketching.
> Specifically, we apply Kernel Density Estimation (KDE) to the pre- and post-sketching embeddings and compute two standard distributional divergence metrics: KL divergence and JS divergence. These metrics measure how similar two distributions are, where smaller values indicate stronger preservation of structural characteristics.
> Results are summarized below:
> | Dataset        | KL Divergence | JS Divergence |
> |----------------|---------------|---------------|
> | Bitcoin-alpha  | 0.0143        | 0.0027        |
> | Bitcoin-OTC    | 0.0062        | 0.0007        |
> | Slashdot       | 0.0003        | 0.0001        |
> | Epinions       | 0.0048        | 0.0003        |
>
> These consistently low divergence values demonstrate that the proposed method preserves the overall distribution and structural patterns of edge features after sketching. This further supports the effectiveness of our edge embedding approach in retaining meaningful information from the original data.
>
> *Q3: error bound ... empirical study*
>
> A3: We conduct an empirical study to evaluate how well the sketch-based estimator approximates the true edge similarity under different deviation thresholds $\epsilon$. Specifically, using the Epinions dataset as a representative benchmark for signed networks, we compute the probability that the deviation between the estimated similarity and the true similarity exceeds a given $\epsilon$, and compare it with the theoretical upper bound $2exp⁡(−2D\epsilon^2)$.
> | $\epsilon$     | the probability of deviation | upper bound |
> |-------|-------------------------------|-------------|
> | 0.040  | 0.49941                       | 0.52729     |
> | 0.0425| 0.47720                       | 0.48794     |
> | 0.045  | 0.45499                       | 0.44859     |
> | 0.0475| 0.43377                       | 0.40824     |
> | 0.050  | 0.41255                       | 0.36788     |
>
> The empirical deviation probabilities closely follow the theoretical bound. This demonstrates that the error between the estimated and true similarities remains tightly concentrated, providing empirical support for the theoretical guarantee of our estimator.

---

> > ### Comment · Reviewer_dXe3 · 2025-08-03
> >
> > The authors have adequately addressed all of my concerns in their rebuttal. I am satisfied with their clarifications and maintain my positive evaluation. I recommend acceptance of the manuscript.

---

> > > ### Author Response · Authors · 2025-08-06
> > >
> > > Thank you for your great support!

---

### Official Review · Reviewer_9SJs · 2025-06-30

**Clarity:** 4
**Significance:** 3
**Originality:** 4
**Rating:** 5
**Confidence:** 4

**Summary:**

Graph neural networks are a popular modeling paradigm for graph structured data, and have enjoyed popularity in both research and industrial settings for exactly this purpose. One common application of GNNs is the construction of node-embeddings that can be used in downstream applications through transfer learning. While popular, this approach traditionally suffers from two different problems -- poor computational scaling and limited expressivity. This is usually done by computing embeddings, and is usually done in a self-supervised fashion. It is well known at this point that using equivariant GNNs to do this at a node-level leads to the isomorphic node problem. For signed graphs, many methods further depend on balance theory which, when violated, can lead to poor results. The authors situate their work in this area of the literature, and present EdgeSketch+. This method presents a computationally efficient technique for generating edge-embeddings, with a strong focus specifically on signed graphs. Efficiency is realized through the use of locality sensitive hashing, which is done in a way that captures higher-order simplical structures. The authors explore the theoretical aspects of their work and contribute multiple interesting bounds, and then present experiments on link sign prediction intended to justify their method.

**Questions:**

1. How would you extend this to higher-order simplical complexes?
2. Are there any modifications that you'd make to the method to make it inductive? It seems like none would be required, but I wanted to make sure.
3. How would you make the model more expressive?
4. Would a more advanced decoder yield better results?
5. Could this be applied to non-signed link prediction?
6. How does computational performance change as edge density increases?
7. How sensitive is the method to the choice of distance metric in the LSH family?

**Ethical Concerns:**

["NO or VERY MINOR ethics concerns only"]

**Final Justification:**

The authors provided clear answers to my questions and have further improved their work. Initially I viewed this paper as a clear accept and this is only more true with the improvements.

**Limitations:**

yes

**Quality:**

3

**Strengths And Weaknesses:**

**Strengths**
1. The work is well motivated and addresses a significant problem in the space.
2. The work is well written and presented. There are no significant typos, and is relatively easy to follow despite the mathematically intricate nature of the work.
3. Figure 1 provides a fantastic intuition for the problem that sets the tone for the rest of the work
4. 4.3, the incorporation into linear learning models is just _cool_.
5. Runtime performance validates that this model just works and is just fast. 546x speedup is incredible.
6. The hyperparameter sensitivity is clearly experimented and explored, and the method looks fairly robust
7. The theoretical results are clearly outlined and succinct. In particular, the concentration bound is particularly clear.
8. The ablation studies clearly demonstrate the importance of each component

**Weaknesses**
1. The experiments also lack standard deviations
2. Only 4 datasets
3. Limited to 2-simplicies without a path forward towards expanding to higher-order tasks
4. The application is quite limited in that it's focused only on signed networks with strong triangular structures.
5. Missing baselines, although I'm happy to give them a pass given that the code isn't publicly available.

---

> ### Author Rebuttal · Authors · 2025-07-31
>
> *Q1: extend to higher-order simplicial complexes*
>
> A1: We extract higher-order simplicial complexes from the graph, and then construct the corresponding adjacency matrices and the corresponding Hodge Laplacians  to describe the adjacency relationships between simplices.
>
> *Q2: any modifications ... to make it inductive?... It seems like none would be required, but I wanted to make sure*
>
> A2: The model does not require specific modifications. If encountering more complicated graphs that need to be extended to higher dimensions, similar to dimensions 0, 1, and 2, we will construct the corresponding boundary adjacency matrix  $\mathrm{B}$ and compute Hodge Laplacian  $\mathrm{L}$.
>
> *Q3: How ... more expressive*
>
> A3: To enhance the model's expressiveness, we could explore extending the dimensionality of simplices, incorporating edges into higher-order simplicial structures, or integrating additional contextual features, such as edge weights or temporal dynamics, to enrich its representational capacity. In future work, we will further explore to improve the expression ability.
>
> *Q4: more adavenced decoder?*
>
> A4: We conduct an experiment employing a Multilayer Perceptron (MLP). Although this approach is more time-consuming, the classification results are slightly better than those of logistic regression, as shown in the table below.
>
>  Bitcoin-alpha
> | Metric    | Paper  | MLP |
> | --------- | ------ | ------- |
> | Binary-F1 | 0.9780 | 0.9791  |
> | Accuracy  | 0.9581 | 0.9605 |
> | AUC       | 0.8829 | 0.9175 |
> | Macro-F1  | 0.7687 | 0.8110 |
> | Runtime(s) | 4.03 | 40.10 |
>
> Slashdot
> | Metric    | Paper  | MLP |
> | --------- | ------ | ------- |
> | Binary-F1 | 0.9209 | 0.9223  |
> | Accuracy  | 0.8729 | 0.8786  |
> | AUC       | 0.8914 | 0.9223  |
> | Macro-F1  | 0.7995 | 0.8222  |
> | Runtime(s) | 140.44 | 1238.07 |
>
> *Q5: applied to non-signed link prediction?*
>
> A5: The goal of non-signed link prediction is to predict whether an edge exists between two nodes in an unsigned network. By contrast, signed link prediction aims to predict the signs of **existing** edges  in a signed network, representing a fundamentally different task. Our method for signed networks relies on existing network structures to generate edge representations, which currently has limited applicability to non-signed link prediction. In future work, we will further explore to find suitable solutions.
>
> *Q6: computational performance change .... edge density increases*
>
> A6: In Section 5.3 Scalability, we test the time consumption as edge density increases with a fixed number of nodes. The experimental results show that the time displays slow quadratic growth.
>
> *Q7: distance metric in LSH*
>
> A7: In Section 5.6, we evaluate the effects of two different distance metrics on the results. Combined with the theoretical analysis in Section 4.2, we conclude that the Hamming distance is a suitable distance metric for the LSH-based embedding method.

---

> > ### Comment · Reviewer_9SJs · 2025-07-31
> >
> > I thank the authors for their thorough response to my questions and identified weaknesses. My concerns are addressed. Thank you.

---

> > > ### Author Response · Authors · 2025-08-01
> > >
> > > Thank you for your great support.

---

### Official Review · Reviewer_uWUZ · 2025-07-01

**Clarity:** 4
**Significance:** 3
**Originality:** 3
**Rating:** 6
**Confidence:** 4

**Summary:**

This paper points out the limitations of node-centric graph embedding methods for link sign prediction, and proposes to overcome the limitations by introducing higher-order interactions in the form of simplicial complexes. It proposes a learning-free method EdgeSketch+, which computes edge embeddings iteratively by performing locality-sensitive hashing on newly defined edge surroundings. Theoretical analysis is provided to characterize the key properties of EdgeSketch+. Experimental results on four public datasets validate the effectiveness of EdgeSketch+. Extensive ablation studies help to understand the characteristics and behaviors of EdgeSketch+, e.g., the scalability, and the sensitivity to the hyperparameters.

**Questions:**

Q1. To my understanding, triangles play special roles in link sign prediction. There are multiple possible configurations of a triangle, which present different real-world meanings and have different impacts on link sign prediction. Is it possible to relate your methods to such observations? For example, you may build a connection with the traditional theory of signed networks, provide statistics on different configurations of triangles, or provide a case study on how these triangles improve the accuracy of link prediction.

Q2. Can you discuss how EdgeSketch+ can deal with dynamic signed networks? For example, how to update the simplex embeddings and which are relevant? What is the time complexity and space complexity?

**Ethical Concerns:**

["NO or VERY MINOR ethics concerns only"]

**Final Justification:**

The rebuttal from the authors has addressed my concerns. Specifically, their answer to Q1 enhances the explainability of EdgeSketch+, and the answer to Q2 precisely describes how EdgeSketch+ can be extended to dynamic signed networks.

**Limitations:**

Yes

**Quality:**

3

**Strengths And Weaknesses:**

**Strengths**

S1. This paper is clearly and concisely written. It is easy to follow and understand.

S2. It is novel to explore simplicial complexes for link sign prediction. The choice of LSH achieves a good balance between accuracy and efficiency, which makes EdgeSketch+ practical for large-scale real-world datasets.

S3. The theoretical analysis characterizes the key properties of EdgeSketch+, which help to understand the behavior of the algorithm.

S4. The experimental design is comprehensive, and the results are significant in most cases. Specifically, the results on the two large datasets, Slash. and Epin., validate the superior performance of EdgeSketch and its computational efficiency.

**Weaknesses**

Currently, I am not aware of any major weaknesses in the paper.

---

> ### Author Rebuttal · Authors · 2025-07-31
>
> *Q1: multiple configurations of a triangle ... statistics on different configurations of triangles*
>
> A1: Different triangle types — such as (+,+,+), (+,+,-) (+,-,-) and (-,-,-) configurations — have distinct real-world interpretations under social balance theory, and potentially different impacts on link sign prediction. To empirically explore this, we conduct an ablation study where we selectively exclude specific types of triangles. Specifically, we produce four ablation variants on  Bitcoin-OTC by removing one type of  (+,+,+), (+,+,-) (+,-,-) and (-,-,-) during simplex extraction.
> | Metric | Paper | w/o (-,-,-) | w/o (+,-,-) | w/o (+,+,-) | w/o (+,+,+) |
> | --------- | ------ | ------- | ------- | ------- | ------- |
> | Binary-F1 | 0.9681 | 0.9606  | 0.9569  | 0.9615  | 0.9512  |
> | Accuracy  | 0.9413 | 0.9361  | 0.9314  | 0.9376  | 0.9342  |
> | AUC       | 0.8695 | 0.8578  | 0.8567  | 0.8594  | 0.8564  |
> | Macro-F1  | 0.8137 | 0.8098  | 0.8006  | 0.8096  | 0.8048  |
>
> As shown in the updated results, excluding different triangle types generally leads to noticeable performance drops, with the (+,-,-) and (+,+,+) configurations contributing more significantly to prediction quality. This observation aligns with social balance theory, where (+,+,+) and (+,-,-) are regarded as balanced configurations, indicating structurally stable and consistent relationships. The fact that removing these balanced triangles leads to larger performance degradation suggests that such structures carry stronger and more reliable predictive signals in signed networks. In contrast, unbalanced configurations (e.g., (+,+,-) and (-,-,-)) are often transient or noisy, contributing less decisively to link sign prediction. These results highlight that EdgeSketch+ effectively captures higher-order signed structures and exploits structurally meaningful patterns aligned with established social theories.
>
>
> *Q2: dynamic signed networks*
>
> A2:
>  EdgeSketch+ is naturally suited for dynamic signed networks due to its non-learning and localized design. When a new edge is added or removed, only its surrounding simplicial structures—specifically, the affected 1- and 2-simplexes—need to be updated. For example, when a new edge connects two nodes, we identify newly formed triangles (2-simplexes) with shared neighbors and update the relevant boundary matrices accordingly. Embedding regeneration is then performed only for the modified edge and its neighboring simplices, using the same pre-sampled LSH projections to ensure consistency. This design avoids recomputing embeddings for the entire graph.
>
> For a static graph, the overall time complexity is $O(N_1^2 + \bar{v} N_1D^2R)$, where $N_1$ is the number of 1-simplices (edges), $\bar{v}$ is the average node degree, $D$ is the embedding dimension, and $R$ is the number of hash rounds. This complexity covers the extraction of simplicial complexes, construction of boundary and Laplacian matrices (e.g., $\mathrm{B}_1$, $\mathrm{B}_2$, and $\mathrm{L}_1$), and multiple-round random projections. The corresponding space complexity is $O(N_1D + D^2R)$, including storage for edge embeddings, Hodge Laplacians, and random projection vectors.
>
> For dynamic signed networks, the incremental time complexity is $O(\Delta ED^2R)$, where $\Delta E$ is the number of modified edges. Due to the fixed projection mechanism, many previously computed embeddings can be cached and reused, which further improves runtime and memory efficiency. The space complexity during update remains $O(N_1D + D^2R)$, as it reuses the same embedding structure and projection vectors without duplicating storage.

---

> > ### Comment · Reviewer_uWUZ · 2025-08-02
> >
> > Thanks for your response. Please add the experimental results and the discussion in the future version of your paper. Additionally, I would recommend reporting the numbers of different triangle types in the revised paper, so that the readers can better relate the contribution of different types of triangles to the accuracy of link sign prediction. I will raise my rating to 6.

---

> > > ### Author Response · Authors · 2025-08-02
> > >
> > > Thank you for your appreciation on our work. We will add the supplemented experiments and discussion in the revised version.

---

### Official Review · Reviewer_Lwco · 2025-07-03

**Clarity:** 3
**Significance:** 2
**Originality:** 2
**Rating:** 4
**Confidence:** 3

**Summary:**

This paper introduces EdgeSketch+, a fast and simple signed network sketching algorithm that leverages simplicial complexes and Locality Sensitive Hashing (LSH) to efficiently generate edge embeddings for signed graphs. Instead of learning node embeddings and deriving edge scores, it directly embeds edges by exploiting higher-order simplicial structures (0-, 1-, and 2-simplexes) to capture rich local geometry, while maintaining scalability through randomized hashing. Experimental results on standard signed network datasets demonstrate competitive prediction performance with up to 546× lower runtime than GNN-based baselines, with ablation studies highlighting the importance of simplicial components.

**Questions:**

- Clarify the notation and terminology in Algorithm 1.
- State explicitly the dimensionality of the embedding after each iteration and after aggregation.
- The seed is chosen and is not random. Was it fixed arbitrarily or cherry picked? How sensitive are the results to this choice? Is fixing a single seed standard practice in this line of work? Some discussion of randomness/stability would help.
- The largest signed dataset used (Epinions) is still small by modern standards. Scalability is a known issue with such higher-order methods. Can the authors comment on whether EdgeSketch+ applies to unsigned graphs, and how it might scale and perform on much larger datasets like those in OGB?

**Ethical Concerns:**

["NO or VERY MINOR ethics concerns only"]

**Final Justification:**

The authors have adequately addressed my concerns in their rebuttal, and I am satisfied with the clarifications provided. While the contribution is somewhat incremental, the method is technically sound, efficient, and well-supported by experiments. I will maintain my positive evaluation.

**Limitations:**

Authors have addressed the limitations.

**Quality:**

3

**Strengths And Weaknesses:**

### Strengths
The paper is well-written and easy to understand. EdgeSketch+ is simple yet elegant — combining higher-order simplicial structures with locality-sensitive hashing to capture rich interactions without expensive training. Its ability to use 0-, 1-, and 2-simplexes efficiently while preserving key properties is compelling, and the theoretical guarantees strengthen the contribution. The experiments are thorough and well-designed, showing competitive or better accuracy than strong baselines with much lower runtime. The ablation studies convincingly show the importance of each simplicial component and highlight the scalability and practicality of the approach.

### Weaknesses
While EdgeSketch+ is the first to use simplicial complexes for signed networks, the overall architecture presented has been around for a while.

In Algorithm 1, lines 5 and 6 refer to convolution operators `ConvB1` and `ConvB2`. The two arrays make it look like $L_1$ and $L_2$ are used for convolution. However, the text says they use adjacency matrices $B_1$, $B_2$. Shouldn’t these be better referred to as $ConvL_0$ and $ConvL_2$ in the context of Hodge Laplacians, for consistency? Please clarify notation and what $L_0$, $L_1$, $L_2$ actually mean in this step. Also, the dimension of embeddings after each step is not clear and would be helpful to include them to for ease of reading.

Since the signed edges can be treated edge orientation in a simplicial complex, it would strengthen the empirical evidence if EdgeSketch+ is compared with simplicial representation learning methods like SaNN, SGAT, etc.

---

> ### Author Rebuttal · Authors · 2025-07-31
>
> *Q1: Notation & terminology*
>
> A1: The edge embedding model focuses on the adjacency relationships related to edges in the convolution operation. $B_1$ describes the boundary adjacency relationships between nodes and edges, $B_2$ describes the upper boundary adjacency relationships between triangles and edges. The Hodge Laplacian  $L_1$, computed from  $B_1$ and $B_2$ via Eq. (1), encodes the edge-to-edge adjacency relationships, while $L_0$ and $L_2$ describe the adjacency relationships of nodes and triangles, respectively. Therefore, we use $B_1$ and $B_2$ rather than $L_0$ and $L_2$.
>
> *Q2: dimensionality of embedding*
>
> A2: After each iteration and aggregation, the embedding dimension is 300.
>
> *Q3: seed*
>
> A3: In the EdgeSketch+ code, we use the current system time as the random seed. As mentioned in Appendix E, each experiment is independently repeated for 5 runs with different random seeds for each run, which complies with standard practice.
>
> *Q4: larger dataset in OGB*
>
> A4: We select ogbl-biokg from the OGB dataset, which contains 5,088,434 edges, and generate edge embeddings using  EdgeSketch+. The embedding time is as follows.
> | Dataset     |  Runtime (s) |
> |-------------|-------------|
> | ogbl-biokg  | 6683.48  |
>
> The results demonstrate the scalability of our method.

---

> > ### Comment · Reviewer_Lwco · 2025-08-03
> >
> > Thank you for your response and clarification.
> >
> > **OGB Results**: Thanks for providing the runtime on OGB. Could you also provide the performance metric for the `ogbl-biokg` dataset and indicate how the model ranks on the corresponding leaderboard?
> >
> > **Comparison with Simplicial Baselines**: A comparison with existing simplicial neural network methods would be valuable. If such a comparison is not feasible, it would be helpful to include a clear justification for why that is the case.

---

> > > ### Author Response · Authors · 2025-08-06
> > >
> > > *OGB Results*
> > >
> > > The ogbl-biokg dataset is an unsigned graph with 51 distinct label types, on which we conduct a multi-class classification task. According to the original paper, RSGNN [1], SDGNN [3], SGCL [6], and edge2vec [17]  run out of time, and EdgeRWSE [26] and MPSketch [37] run out of memory on the Slashdot and Epinions datasets with no more than a million edges. Additionally, SNEA’s [4] implementation depends on edge signs, making it incompatible, and SiGAT [7] runs out of time on ogbl-biokg. Consequently, we only report the results in comparison with GSGNN+SGA [2,54] and TER [18]. We use accuracy as the evaluation metric for multi-class classification, with the results presented below.
> > >
> > > | Method     | Accuracy | Runtime (s) |
> > > |--------------|----------|-------------|
> > > | EdgeSketch+   | 0.6483   | 9876.68     |
> > > | TER           | 0.3936   | 4045.97     |
> > > | GSGNN+SGA     | 0.5094   | 17195.99    |
> > >
> > > *Comparison with Simplicial Baselines*
> > >
> > > We have compared with EdgeRWSE [26], which can represent each edge in the neural network framework with simplicial complexes. The experimental results in the original paper show that our proposed method outperforms EdgeRWSE in terms of performance and runtime. SCN [28], SGAT [24] and SaNN [23] embed edges, nodes and the whole graphs in the message passing networks with simplicial complexes. Unfortunately, the three simplical neural network methods have not releassed their codes.

---

> > > > ### Comment · Reviewer_Lwco · 2025-08-06
> > > >
> > > > Thank you for your response and clarification.
> > > >
> > > > **OGB Results**: Thanks for sharing the OGB results. Why did the authors choose to report accuracy? I noticed that the OGB leaderboard for `ogbl-biokg` has MRR as the metric used. Can the authors also share the MRR for their model to show they compare on the leaderboard? Also, there seems to be some discrepancies between runtime from your original rebuttal and this response. Please explain that as well.
> > > >
> > > > **Comparison with Simplicial Baselines**: I believe `geometric-intelligence/TopoBench` has a few implementations for simplicial models (SCNN, SCN, SAN). The authors can include these baselines in the revised manuscript.

---

> ### Author Response · Authors · 2025-08-09
>
> *OGB Results*
>
> Thank you for the suggestion. We select ogbl-biokg from the OGB benchmark to provide a large-scale, real-world graph scenario. Since our task is formulated as edge classification, we report Accuracy, which directly measures the correctness of edge classfication. The ogbl-biokg leaderboard focuses on knowledge graph completion, which involves predicting missing entities in triplets (either (entity, relation, ?) or (?, relation, entity)) in terms of MRR; by contrast, our method predicts edge signs/labels by generating edge representations, which does not generate tail predictions for query triples as required by the task definition. Therefore, these two tasks are fundamentally different in nature, making our approach unsuitable for  knowledge graph completion. Regarding the runtime discrepancy, the previously reported 6683.48s corresponds to the time required for generating the edge embeddings, while the 9876.68s includes both embedding and downstream classification.
>
> *Comparison with Simplicial Baselines*
>
> We have carefully reviewed the TopoBench framework and the corresponding paper [1]. As stated in Section 5.1 of the TopoBench paper,
> > "Four types of tasks are considered: node classification (seven datasets), node regression (seven datasets), graph classification (seven datasets), and graph regression (one dataset)."
>
> None of them includes edge classification tasks. Moreover, after inspecting the available codebase, we confirm that it does not provide implementations for edge embedding or edge-level tasks, which are central to our setting. As a result, we are currently unable to run these methods for a fair comparison on our task.
>
> [1]Telyatnikov L, Bernardez G, Montagna M, et al. Topobench: A framework for benchmarking topological deep learning[J]. arXiv preprint arXiv:2406.06642, 2024.

---

### Decision · Program_Chairs · 2025-09-17

**Decision:**

Accept (poster)

**Comment:**

The paper introduces EdgeSketch+, a model to learn edge embeddings in signed graphs, balancing accuracy and efficiency. The main idea of the model is to represent edges through an LSH technique, capturing higher-order information of the underlying simplicial complex. The work has several clear merits highlighted by the reviewers, including the novelty of the methodology, the good experimental results, and the theoretical support. During the discussion period, the authors provided clarifications and addressed several of the reviewers' concerns. Overall, the paper makes a solid and well-motivated contribution.